# Exploring Perceived Barriers to Physical Activity in Korean Older Patients with Hypertension: Photovoice Inquiry

**DOI:** 10.3390/ijerph192114020

**Published:** 2022-10-27

**Authors:** Gun-Young Lee, Kyung-O Kim, Jae-Hyeong Ryu, Sun-Hee Park, Hae-Ryong Chung, Marcia Butler

**Affiliations:** 1Department of Gerokinesiology, Kyungil University, Kyungsan 38428, Korea; 2Chungbuk Boeun Naebuk Public Health Center, Boen 28917, Korea; 3Social Welfare Center, Daegu 41052, Korea; 4Health and Fitness Management, College of Health, Clayton State University, Morrow, GA 30260, USA; 5Health Care Management, College of Health, Clayton State University, Morrow, GA 30260, USA

**Keywords:** older adults, hypertension, barriers to physical activity, photovoice

## Abstract

This study attempted to explore the barriers to physical activity of older patients with Hypertension. It aimed to provide robust evidence produced through their eyes. First, through the data analysis of the accelerometer and the decision of the research team, 10 out of the 30 applicants were invited to participate in a photovoice study. Photovoice is one example of participatory action research. Photovoice participants can communicate their unique experiences through photographs, providing a highly realistic and authentic perspective that is not possible to be understood with traditional qualitative research. This study inductively identified four main themes; health illiteracy, distortion of health information, fear of physical activity, and rejection of any life changes. Based on a specific understanding of the population’s perception of physical activity, this study attempted to provide evidence of why many elderly Korean patients with Hypertension stay inactive.

## 1. Introduction

In Korea, the prevalence of hypertension among older adults aged 70 years and older is overwhelming. Among people over the age of 70 years, 59.5% of men and 72.4% of women have Hypertension [1,2]. Hypertension is twice as high as the prevalence of diabetes, which is the second most common chronic disease among the elderly population in Korea. In 2022, a new guideline for assessing high blood pressure was changed to 130/80 mmHg. People with blood pressure at this standard are now categorized as a high-risk group. Korea ranks first in the world in increasing the number of hypertension patients based on the traditional standard of 140/90 mmHg [3]. With the new guidelines of 2021, the number of people with hypertension in Korea has exceeded 12 million [4].

Hypertension is one of the major risk factors associated with death [5]. Hypertension is a causal factor that can affect various functional disorders in the heart, brain, kidneys, eyes, etc., and can eventually lead to diseases such as stroke, chronic kidney disease, myocardial infarction, dementia, necrosis, hypertensive retinopathy, as well as sudden death [6,7]. Moreover, hypertension is known as a silent killer because it is difficult to recognize at an early stage.

There are several treatment modalities for hypertension that have been studied to assess their effectiveness. These include diet/nutritional counseling, smoking cessation programs, stress management [8], and the use of medication [9]. When hypertension is diagnosed, medication is prescribed [4]. 

One of the characteristics of older adults with hypertension is that they have high systolic blood pressure. As the age increases, the systolic blood pressure gradually increases, and after a certain age, the diastolic blood pressure tends to decrease. This is because the elasticity of blood vessels is weakened and stiffened with aging. However, older adults with hypertension can obtain sufficient benefits from regular physical activity, which reduces social costs. 

Several previous studies have shown that physical activity has a positive effect on hypertension [10,11]. Current research also shows that the amount of physical activity has a positive effect on people diagnosed with hypertension, in addition to the use of medication. In other words, physical activity is one of the most effective interventions for patients with hypertension [1,12]. Other studies have shown that physical activity can reduce or eliminate the use of medication by controlling blood pressure levels as the cardiovascular system improves [13,14,15].

The American College of Sports Medicine (ACSM), World Health Organization, and National Institutes of Health provide physical activity guidelines for people with hypertension. Specifically, ACSM recommendations include that elderly patients with hypertension engage in aerobic exercise daily for at least 30–60 min at a moderate intensity level and resistance training/exercises 2–3 times a week. However, some studies have shown that low-intensity physical activity may also have a positive effect on hypertension [16,17]. Research studies from Korea have similarly reported that regular physical activity reduced the total cost of hypertension treatment by 30% [18]. Participation in physical activity can have a positive effect in terms of personal and social costs. Physical activity should be an essential treatment intervention for the older population with hypertension [18,19]. 

In Korea, several problems have been identified as impediments towards physical activity as an alternative or complementary treatment for hypertension. One of the first problems is the lack of consistent guidelines for physical activity among Koreans diagnosed with hypertension. Many countries have published physical activity guidelines for people with hypertension. However, it is difficult to find these guidelines in Korea and there are differences between these guidelines. The absence of specific and consistent guidelines for healthcare providers and the absence of healthcare providers who recommend them are negative factors for physical activity in older adults with hypertension [20]. Treatment of the elderly Korean population with hypertension should involve an interdisciplinary, collaborative approach that includes patients, healthcare providers, and exercise professionals. This interdisciplinary treatment approach can significantly improve the lifestyle and pharmaceutical adherence of this population, as well as improve overall health and reduce social costs [13]. 

The second problem identified is the lack of physical activity or exercise among the older population in Korea, especially those diagnosed with hypertension. In Korea, the older population is generally physically inactive. It is reported that about 77% of older adults do not participate in regular physical activity at least once a week [21]. The Korea Disease Control and Prevention Agency (KDCPA) reported that the walking practice rate for hypertensive patients in Korea over the age of 30 was about 38%, and similar figures were also observed in the area our research team studied. It can be reasonably estimated that the walking practice rate of hypertensive patients aged 70–75 years, the subject of this study, is much lower. Moreover, this fact is the main basis for inferring that the proportion of hypertensive patients aged 70–75 who are practicing a moderate-intensity level of physical activity suggested by ACSM is very low [2]. Physical activity is one of the most cost-effective interventions for older adults with hypertension, but when considering the physical activity level of the general older population, it can be reasonably inferred that older adults with hypertension may not actively participate in physical activity.

The third problem in Korean society can be the increase in the burden on the National Health Insurance Service (NHIS). In the case of Korea, where the NHIS has developed, the NHIS covers most of the cost of primary medication treatment for hypertension, which dominates the prevalence rate. The payment of such medical insurance is currently a huge burden on society [16]. Similarly, in North America, hypertension is the most costly of all cardiovascular diseases, with an estimated direct cost of $69.9 billion in 2010. In addition, in 2015, the indirect cost of hypertension was $27.2 billion, accounting for 23% of the total cost, and the direct cost was $91.4 billion [17]. 

Previous studies on the treatment and management of older hypertensive patients’ physical activity include intervention or experimental studies. Several studies looked at the utilization of various physical activities to reduce hypertension [20,21], financial costs [22,23], pharmacological and non-pharmacological treatment strategies as well as changes in dietary or nutritional intake [24,25], and various medical treatments including nutrition intake [26,27,28,29,30,31,32,33,34,35,36]. The problems of the existing studies so far are as follows: (1) There has been very little research examining objective data on the amount of physical activity in the “daily life” of older adults with hypertension, (2) Currently, there are participatory action research that analyzes the meaning or obstructive factors of physical activity from the perspective of the patient, rather than the researcher. Participatory action research utilizing photovoice has a relatively high possibility to reveal causes of the elderly population’s physical inactivity compared to traditional research methods [37]. In other words, previous research has failed to prove the delicate causes of low levels of physical activity in older adults with hypertension. 

This study will utilize community participatory research with photovoice to investigate why Korean older adults with hypertension are not likely to participate in physical activity. This study will focus on understanding the emotional and other barriers to engaging in daily physical activity to reduce hypertension.

## 2. Method

### 2.1. Study Design

Figure 1 below briefly shows the flow of this study. This qualitative study can be seen as a participatory study in the sense that participants are directly involved in the research process through photovoice techniques. This study also corresponds to a case study in the philosophical aspect of qualitative research [38] because it explored older hypertensive patients’ physical inactivity [39,40]. This study was prepared based on the Consolidated criteria for reporting qualitative research (COREQ), a guideline for qualitative research. Unlike the previous study [21], where we relied on the study subjects’ statements (i.e., patients with panic disorder), we added the use of the accelerometer to assess physical activity instead of relying on the study subjects’ assessment of their physical activity levels (Table 1).

Therefore, in this study, ten subjects with low physical activity were selected based on objective decisions through the accelerometer. To this end, 30 older adults with hypertension first participated in the accelerometer measurement process. Afterward, to objectively check the amount of physical activity and to exclude the possibility of lowering physical activity due to another chronic disease, the co-researchers who are doctors, exercise physiologists, and aging research experts, held a meeting to select the final ten. Since the purpose of this process was to screen candidates to select ten suitable people to perform photovoice, it can be said that the work was conducted to improve the completeness of the photovoice research (Figure 1). 

### 2.2. Keep as Participant Selection

We recruited 47 potential research participants through purposeful sampling from November 2021 to March 2022. All research volunteers filled out the questionnaires regarding demographic data, and the ACSM 2020 PAR-Q+ questionnaire was modified by Dr. Kim to check for chronic diseases. Among the 47 volunteers, 17 patients who had physical activity restrictions due to chronic diseases other than hypertension (e.g., depression) were excluded from participating in the accelerometer study process. Thirty participants were invited to the accelerometer process. For inclusion in the study, the co-researchers reviewed the 30 participants’ accelerometer data to objectively assess the amount of physical activity, or lack thereof. Exclusion criteria included lower physical activity due to other chronic diseases. Ten people who met the criteria were invited to participate in the photovoice research. Their general characteristics are shown in Table 2 below. 

After that, ten people were invited to the photovoice research through a meeting of experts. All participants in this study were patients with hypertension living in Daegu, one of the largest cities in Korea. Participants in this study met the following criteria: (a) patient with hypertension, (b) older adult aged 70 to 75 years of Korean nationality, (c) a person with no problems with communication, (d) a person who does not meet the ACSM hypertension patient physical activity standard and (e) a person with no restrictions in physical activity due to other clinical diseases or musculoskeletal dysfunction (d was determined by M.D. Ryu). For reference, the reason why the participants of this study were operationally limited to those aged 70 to 75 was to minimize errors in the loss of physical activity due to natural aging (Extensive studies of patients aged 75 years and older are likely to cause large errors in quantitative values and subjective perceptions of physical activity). In addition, the number of participants considered suitable for the photovoice study is known to be between 7 and 15 individuals [37]. 

### 2.3. Research Team and Reflexivity

The research team consisted of two qualitative research experts (1 male and 1 female), one exercise physiologist (1 male), a psychological counselor (1 male), a social welfare field specialist (1 female), and a medical doctor (1 male) who want to understand the various problems experienced by research participants. Based on their own specialized experiences, the research team has rich knowledge and understanding of data collection and analysis for patients with hypertension. To avoid their specializations from becoming a bias during the study, the researchers conducted discussions and interactions in various ways during the research period. Specifically, Dr. Kim conceptualized and organized this study, Dr. Lee (male), a professional counselor, conducted three focus group interviews, and MD. Ryu (male), a medical doctor, accompanied all interviews to minimize the risk posed by interviewing patients. Dr. Kim and Dr. Butler analyzed all the qualitative data. Dr. Chung operated accelerometer data and supervised all the processes between Kinesiologist and the other specialist. Dr. Kim received various training related to qualitative research at the University of Illinois. In particular, Dr. Kim completed his education on the diversity of qualitative research through Dr. Denzin’s class as well as Dr. Greene’s class.

### 2.4. Measurement

In this study, an accelerometer was used to select photovoice participants based on objective evidence of the participant’s physical activity. An accelerometer typically uses a week’s worth of data. It objectifies physical activity at all times, including the study participant’s sedentary time. In addition, it is very accurate compared to the questionnaire technique, which reflects individual subjectivity in objectively understanding the amount of physical activity [41,42]. By using the present photovoice, participants can share their unique experiences through photographs, providing the study with a very realistic and authentic perspective that cannot be understood with traditional qualitative research [40]. In this study, the accelerometer is meaningful in accurately recruiting the research participants targeted by photovoice. Therefore, other indicators (ex. Steps, Bout, PA pattern et. al) produced by the accelerometer were not used in this study.

The photovoice study, which is the core of this study, was conducted with the final ten people selected through accelerometer data and expert meetings. In North America, photovoice research has recently expanded to the field of Kinesiology [37]. The present study is based on a protocol developed by Baker and Wang and modified by Kim, which is divided into three main stages: orientation, photovoice implementation, and focus group interview [43].

During the orientation, Dr. Kim, Dr. Lee, and M.D. Ryu attempted to create rapport with the participants and introduced them to all researchers and their achievements. In particular, Ryu, a medical doctor, was asked about their symptoms and medications. Finally, these interactions allowed the researchers and participants to become more trustworthy.

In the photovoice implementation session, participants were trained on the use of disposable cameras, the decision of what photographs to take, and ethical issues related to taking photographs. Every participant received disposable cameras, with an approximate capacity of 27 photographs. The disposable camera was for taking photographs related to perceived problems or barriers to physical activity. After the orientation, participants resumed their daily routines for approximately two weeks and took photographs [21]. In addition, gifts such as terabands and grip balls were given to the older adults who participated even in one research process.

### 2.5. Data Analysis

The accelerometer data was analyzed through Actilife software, and only physical activity data among various indexes were used for the study. 

The photovoice data extracted by the objective amount of physical activity were analyzed with the photovoice analysis and content analysis [21]. Specifically, both researchers and participants in the study have worked on finding some initial codes, themes, theme reviews, entitling their photographs, etc. Researchers have conducted content analysis to discover and classify the statements and themes of interview data produced through focus groups as well as all the research participants’ reviews.

As a result of this process, 93 photos were acquired in total. At the discretion of the researcher, reduplicated photos, and not appropriate printed photos were excluded or important photos were added. The ten participants in this research also got involved in this classification work through pilot interviews and telephone conversations and selected up to four photos they thought were important. In the end, a total of 17 photos were selected that both research participants and researchers agreed were the most meaningful. A focus group interview of these selected photos was conducted with field memos. The SHOWeD technique was employed [37] to lead the focus group interview. The SHOWeD technique is described in Figure 2.

Data collection for focus group interviews was conducted until saturation. In addition, all recordings of the interview were transcribed. After the research was completed, all data on photographs and interviews were discarded. All focus group interviews consisted of two sessions with four participants, one researcher, and one doctor. Each interview lasted between an hour and an hour and a half, with additional interviews being conducted for saturation.

All participants’ statements were recorded and transcribed. The recordings will be destroyed for research ethics after the study is finally published. The interview data were translated from Korean to English with the help of professional translation services. The problematic linguistic expressions based on cultural differences were carefully revised with the expert’s advice. Then, the entire manuscript, as well as the interview data, was modified with the help of editing services. Through these processes, the interviews were translated into English language.

Researchers interpreted the data from focus group interviews and significant statements. The principal researcher arranged the statements with photographs. Then, the research team had several rounds of discussion to derive and refine the statements and themes with photographs [21]. The flow of data analysis is shown in Table 3 below.

### 2.6. Authenticity and Trustworthiness

The present study attempted to increase authenticity and trustworthiness via triangulation, peer debriefing, an independent review panel, thick description, and an audit trail. For triangulation, data such as photographs, statements, and field memos were integrated. There have been several discussion processes to categorize themes for peer debriefings [44]. Further, two independent reviewers will screen the titles of our manuscript and then review the full texts. All researchers attempted to give a rich description, and external experts in the field of qualitative study were asked to audit the process and results of the study, and no particular problems were found. Finally, this study continuously applied the principle of reflexivity to minimize our influence on the participatory study [21,44,45]. 

## 3. Results

### 3.1. Accelerometer

*“The ACSM recommends that individuals with hypertension engage in moderate- intensity (3.2–4.7 MET), aerobic exercise 5–7 d/wk”* [16]

Of the 30 study participants who wore the accelerometer, only 25 succeeded in wearing the accelerometer continuously for a week. The remaining 5 participants were unable to wear the accelerometer continuously for a week, and the data generated from them were excluded from the analysis.

Although there are some studies that very low or low levels of physical activity are effective for hypertension [17], the ACSM guidelines, produced by experts and accumulated evidence, recommend moderate-intensity physical activity for older adults with hypertension. This is to give sufficient advantages to hypertensive patients through physical activities [16]. 7 out of 25 participants practiced moderate-intensity physical activity recommended. This result shows a similar tendency to the walking practice rate of 38.5% of hypertensive patients over the age of 30 in Korea. It was also found that the amount of physical activity naturally decreased with age. The results of this study also showed that less than a third of the older adults participated in moderate-intensity physical activity. The result may indicate that older adults with hypertension do not actively participate in physical activity.

The physical activity data of 25 participants are shown in the table above. Most of the research participants had a high proportion of sedentary life during the past week. Seven of the accelerometer process participants engaged in moderate-intensity physical activity for an average of 30 min or longer, which is the standard of physical activity for hypertensive patients. The rest of the participants did not meet the guidelines of ACSM. None of the applicants exceeded the ACSM’s light activity standard of an average of 1.6 MET per week. All of the light activity shown in Table 4 corresponds to very light activity based on ACSM guidelines for older hypertension patients. Most of the research participants showed the minimum degree of movement to lead a life and generally participated in a moderate-intensity activity at a low rate. Out of a total of 25 participants, 7 participants who exceeded the ACSM’s moderation activity criterion were first excluded from the study. Next, M.D. Ryu and the research team excluded four participants with chronic diseases that could interfere with physical activity in addition to hypertension. Finally, ten photovoice research participants were selected through a meeting of the research team (criterion and purposive sampling). As such, accelerometer data were importantly used to extract older adult hypertensive patients with objectively low physical activity (Table 4).

### 3.2. Photovoice

The photovoice research was analyzed based on the data produced from the research participants’ perspectives. Through the entire focus group interview process, it was found that the Korean elderly were more familiar with the word “exercise” than with the word “physical activity”, so the research team used the word “exercise” to interact with the participants. During the photovoice process, the research team found out that most of the participants, who had stated that they had no chronic diseases other than hypertension, also had other chronic diseases such as low back pain, mild arthritis, and chronic headaches. However, it was also recognized that this did not significantly affect the physical activity of older adults with hypertension. The crucial factors affecting the physical activity of older adults are summarized in the table below. And in this process, microscopic factors that interfere with the physical activity of elderly hypertensive patients were found. In this study, 17 photographs, 9 sub-themes, and 4 main themes emerged (Table 5).

#### 3.2.1. Health Illiteracy

Many studies have reported that physical activity is important for the prevention of hypertension and its complications. However, there was a big difference between the opinion of the academic world and the perception of patients. The older adults who participated in this study were generally unaware of the positive effects of physical activity on hypertension. They had no idea when, how much, or what kind of physical activity was effective. In addition, they were completely unaware of the existence of physical activity guidelines for hypertension.

##### Medication Dependency

Medication is the most common way to treat hypertension. Although other alternative methods have a positive effect on lowering blood pressure such as lifestyle, nutrition, and physical activity. However, some older participants in this study showed blind faith in using medication. The participant below directly shows the belief about medication through the picture of the blood pressure gauge.

The following research participant also made a similar statement with Figure 3.


*Hypertension is nothing compared to other diseases…We only need to take one pill a day. Take your medicine and check your blood pressure! My blood pressure is completely normal. What are you worried about? Worrying too much is a worse habit… (I)*


The following research participant also made a similar statement with Figure 4.


*I take my medicine well…I also drank the onion juice (healthy food) that my children gave to me… No problem. I take medicine and my blood pressure is under control… Except for high blood pressure, I am healthy. I am not ill with a fatal disease. Don’t worry. (J)*


Hypertension is a very dangerous factor that can damage the blood vessels of the brain, heart, and kidneys, and it can cause various complications such as stroke, myocardial infarction, heart failure, renal failure, arteriosclerosis, and certain cancers [45,46,47,48]. However, most of the older adults who participated in this study were not aware of this.

##### A Negative Reference Group

The older adults who participated in this photovoice study were negatively affected in physical activity due to a negative reference group (physically inactive older adults around them). According to M.S. Park, who works at a senior welfare center, there are a lot of older people over the age of 75 here, and most of them have hypertension. These older adults become a physically inactive reference group. In addition, it was observed that most of the older adults in the center preferred programs related to recreational activity over physical activity. In other words, it is natural for the study participants to see that people around them have hypertension but only take medicine and do their preferred activities without any intervention or care. In the center, individuals who manage hypertension with special care are rare, which can also be a negative cause of participation in physical activity (Figure 5).


*Take a look here and there Does anyone exercise? Hypertension is not even a disease anymore. You only need to take one pill … No one works out. No one is exercising. If I exercise alone here, other people will look at me strangely. (B)*


The following study participant showing Figure 4 was willing to participate in physical activity but did not because of peer pressure at the welfare center (Figure 6).


*I know that exercise is good for everything, not just for hypertension. I also want to do it sometimes. But everybody plays like that (Figure 6), so I hang out too. Is it fun to exercise alone? (C)*


##### Misperception of Appropriate Physical Activity

Some older adults who participated in this study thought that they were physically active. This indicates the difference between subjectivity and objectivity for physical activity. If this study did not employ the use of the accelerometer, it is assumed that this difference would have been overlooked. (Figure 7).


*I don’t exercise like this. Is this machine (accelerometer) broken? I’ve also tried a pedometer, but it is not accurate. I cannot believe this (result)…I cannot understand. (A)*


The above study participant had a strong belief in his subjective cognition that he could not even trust the accuracy of the accelerometer. The following study participant also talked similarly about his exercise participation with Figure 8.


*I also play table tennis and walk… I exercise a lot, but the machine is weird. Is this machine accurate? No matter how I think about it, it’s weird… (D)*


M.S. Park, who is well acquainted with this study participant, explains that the participant usually does exercise for a while and then quits because it is hard. This participant also plays table tennis, but after a while, he usually sits and rests. It is noteworthy that the amount of physical activity most objectively measured with an accelerometer does not match the amount of physical activity subjectively perceived by hypertensive patients. 

#### 3.2.2. Distortion of Health Information 

Unlike Health Literacy, the older adults who participated in this study reinterpreted unspecific information on their own or received distorted information from those around them. Therefore, there were cases in which they could not do physical activity even if they were willing to do or it were unable to perform the physical activity due to incorrect information.

##### Inappropriate Health Information from M.D.

The participant in the study below did not understand the words “moderate exercise” often referred to by doctors when they prescribe medication. This is not a problem for the study participant. It may be possible that Korean physicians may not be aware that patients do not understand what is meant by moderate physical activity. Therefore, they may assume that patients know what moderate physical activity entails and do not seek patient’s understanding (Figure 9).


*Every time I go to the hospital, the doctor says, ‘Take your medicine and exercise moderately’, but I don’t know what he means by moderate exercise. So I just walk around town from time to time on my terms. The doctor told me to exercise moderately, and after I was diagnosed with hypertension, I stopped going to my favorite gate ball. When I hit the ball, I have to use my strength… I wonder if I’m walking too much while playing gateball… (B)*


The study participant below believed that he had the will to do the physical activity but believed the wrong information that people with hypertension could not do it because their blood pressure would rise if they did the physical activity (Figure 10).


*Figure 10 shows the playground…I know that exercise is good…But I have hypertension… When hypertensive patients exercise… It is said that exercise makes blood pressure increase and it’s dangerous.… But what kind of exercise can I do? I also want to do my favorite exercise and go to the sauna if I don’t have hypertension. (F)*


Regular physical activity by itself lowers blood pressure [49]. Although physical activity has a positive effect on blood pressure, regardless of medication, the older adults in this study did not participate in physical activity due to the acquisition of insufficient and distorted information. Some of this misinformation may be due to physicians not providing accurate information regarding physical activity and its effects on reducing blood pressure.

##### Lack of Expertise in the Organization

This study population of older adults attended a senior welfare center. There is no obligation for nurses or doctors to provide services at senior welfare centers in Korea. It is also not obligatory to hire a professional who can guide physical activity. Therefore, most of the systems in the center are operated by social workers. In this regard, it is politically favorable for the senior welfare center not to engage in excessive physical activity. In addition, it is not easy to obtain related information. This has an important influence on the wrong choice of physical activity among older adults.

The following study participant made the statement with Figure 11.


*Is there anyone here who is not sick at the welfare center? Each one hurts. But who teaches us what to do?… There is an exercise program. But how do I know if it’s right for me or not? I’m not an expert… Then I just sit down, watch TV, talk, and play with people. (C)*


As a result of analyzing the physical activity programs of several senior welfare centers, the research team found that most of them required the lowest level of physical movement. For example, the most provided exercise was the one for flexibility and balance. A more serious problem is that the proportion of physical activity programs in the total programs provided by senior welfare centers is very low. This may be due to the current physical activity programs that are organized with a focus on preventing accidents or injuries that may occur during physical activity, rather than focusing on individualized physical activity programs.

One participant showed Figure 12 and stated as follows:


*There are so many people here, so we have to choose the program by ourselves. I have hypertension, but who makes a program for hypertensive patients? No… Still, my friends know that I have hypertension, so they tell me not to take some programs (physical activity)… (G)*


This participant was accepting the reality of insufficient and distorted information from neighbors or friends quite naturally. A problem with the institution affects the older adults individually.

#### 3.2.3. Fear of Physical Activity

##### Trauma or the Fear Experienced

Some of the research participants had negative feelings about physical activity. This is not limited to hypertensive patients, but it is common in older adults. Whether it happened to them or others, some things can be traumatic for older adults.

The following study participant talks about her trauma through Figure 13.


*My son also had hypertension. He often exercise, saying that he could get better when he exercised. But one day, he suddenly collapsed, and from then on, he was hospitalized…After all, I am older than him, what kind of exercise can I do? My son is like that now after he exercised… I cannot. (I)*


The study participant below shares a similar experience.


*My sister had hypertension, and she collapsed from walking too much… So I (with hypertension) also took a computed tomography… From then on, I try not to do anything too hard. (H)*


It is not possible to find out exactly why the son with hypertension suffers from sequelae because we could not get a detailed explanation. We also did not get any details of why the study participant collapsed from walking. However, these incidents were extremely rare cases. As a result, this study participant had trauma from physical activity and for this reason, she refused to engage in physical activity

In the following case, the study participant stopped doing physical activity after experiencing a negative experience. This can be seen as a kind of traumatic reaction (Figure 14).


*I like mountains, so I have been climbing for a long time … I went climbing while taking hypertension medication. But it was a cold day, maybe… I was going up the mountain regardless of the weather. Suddenly I felt dizzy and my legs were losing strength… Other hikers called 911 (See Figure 14). It must have been about two years ago. Since then, I haven’t exercised at all. (E)*


The study participant below also made very similar statements.


*I used to walk often because I heard that exercise was good for hypertension… In the evening… But one day, while walking, my hands started to swell. I was startled, so I quit exercising right away, went home, and took a rest. I haven’t exercised at all since then. I can just take the medicine… For what benefit did I do it… (J)*


Numbness, pain, discomfort, and muscle aches are the side effects that hypertensive patients can have after participating in physical activity [49], and various side effects can appear when exercising at an appropriate level or higher. However, physical changes caused by physical activity can be a great fear for older adults when they do not know these facts.

##### Anticipatory Anxiety

The following study participant was unable to participate in physical activity due to anticipatory anxiety (Figure 15).


*It would be fine if I exercised with my friends (See Figure 15) at my age… <omitted>… If I fall while exercising alone, who can save me? I can’t help but die. Those who don’t have many friends like me are not good at doing things together. According to the news, an old man died while climbing the stairs and was found a few days later. I also have hypertension… I am scared. (H)*


The following study participant also made similar statements with Figure 16.


*I went out to exercise because I heard that exercise is good for health… <omitted>… I remembered that I had not taken my pills for hypertension in the morning! All of a sudden I got scared… So I went straight home and took some medicine… I still carry my medicine with me all the time. After that, I cannot exercise… There are many times when I am confused as to whether I have taken the medicine or not… I often get confused as to whether I took the medicine or not…(E)*


Due to extremely rare negative experiences or personal mechanisms to induce fear or anxiety, some older adults in this study were avoiding physical activity.

#### 3.2.4. Rejection of Any Changes

##### The Importance of the Current Lifestyle

Finally, some study participants admitted that their current blood pressure was being maintained with medication and spoke as if they had no will to improve (Figure 17). 


*Do I have to exercise, watch my diet, and quit drinking and smoking to live longer at this age? I take medication for hypertension and my blood pressure is being treated well. So I do whatever I want to do. I eat my favorite food and enjoy drinking and smoking. (G)*


As seen in Figure 17, the participant currently drinks and smokes. Drinking and smoking are the main factors that cause blood pressure increase, so hypertensive patients must avoid them. The participant knew this, but he was just satisfied with his current life. He was not aware of the reasons to improve his lifestyle. It can be thought that hypertension, which is called a silent killer, has different characteristics from diseases that alert patients because the symptoms of hypertension are not easily recognized, or there is no particular physical response to the process leading to complications.

##### Lack of Motivation to Improve Their Life

The following study participant showed an unwillingness to improve his lifestyle through Figure 18.


*I’m weak-willed, so no matter what I do, I can’t do it well until the end. Even if I go on a diet, I always fail. I tried to exercise too, but it doesn’t work for me. I just take medicine well… haha I just want to live well… I’ll die when I die… haha. (F)*


One study participant made a similar statement with Figure 19.


*Exercise? How long to live.… I don’t want to get stressed and I want to do what I want to do… Who here doesn’t have hypertension? I want to live happily while I am alive. (I)*


Although the statements of the two study participants appear similar, the details are slightly different. The first participant was satisfied with the present and had a strong attachment to the things she was doing, whereas the second participant had an easygoing personality and did not have great greed in all areas, including physical activity, due to his weak will.

Hypertension is very common, but it is also a very scary disease. However, the above research participants seemed to regard hypertension as having a cold. Both participants appeared so calm and carefree that the researcher was surprised that they were not worried about their hypertension.

## 4. Discussion


*“Why more Korean older adults with hypertension are physically inactive?”*


This study challenged to verify the microscopic cause of “Why do older adults with hypertension not participate in physical activity?” based on objective data from accelerometers that the majority of them do not engage in physical activity to improve hypertension. The final 10 participants who were physically inactive participated in the photovoice study. They repeatedly explored the unique interfering factors of their physical activity together with the research team until theoretical saturation was reached. As a result, this study discovered four decisive main themes: Health Illiteracy, Distortion of Health Information, Fear of Physical Activity, and Rejection of any changes.

### 4.1. Health Illiteracy

The first sub-theme related to this is “Medication Addition”. Although all study participants are taking medications for hypertension, hypertension is a terrifying chronic disease that requires overall well-controlled lifestyle habits [50]. However, some older adults in this study had a strong belief that medication would solve everything. Therefore, no reason to do physical activity daily. To them, hypertension was perceived as a mild chronic disease requiring only medication. This greatly increased the possibility of other health complications, as mentioned earlier [51]. Second, about “A negative reference group”, some older adults who participated in this study had negative influences from those around them. Due to their peers who also have hypertension and take medication to control it believed that they do not need to engage in physical activity, the study participants may be less likely to engage in physical activity as well. It is well known that peer pressure influences individual behaviors [52]. The study participants were not aware of the severity of having hypertension and did not have the opportunity to lower their dependence on medication, nor have support from their peers. Studies have shown that social support, lifestyle improvement, and physical activity decrease the dependence on the use of medication [53]. The third theme is “Misperception of appropriate physical activity” This was a very significant finding because this study objectively measured physical activity with an accelerometer. The difference between individual objectivity and subjectivity regarding physical activity was verified. Significantly, the meaning of “moderate physical activity” for older adults who are not experts in physical activity can vary greatly from person to person, while guidelines [54] published by most reliable institutions recommend moderate or higher physical activity to get benefits from physical activity. The lack of understanding of the level of physical activity is similarly reported in a Polish study [51]. 

### 4.2. Distortion of Health Information

The first sub-theme is “Inappropriate health information from M.D.” According to the statements of the study participants, doctors recommend “moderate exercise” while prescribing medication every month. However, it is more difficult for patients to estimate the appropriate physical activity for them, as they do not know how much moderate-intensity physical activity means because they are not a physical education expert. M.D.’s inadequate patient education has already been reported in research [55]. M.D.’s must provide older patients with hypertension with recommendations for cost-effective lifestyle interventions and individualized recommendations for physical activity as well as medication treatment. However, according to the statements of the study participants in this study, most doctors give counseling within 5 minutes, prescribe medication, and repeatedly talk about “appropriate physical activity and diet control”without providing specifics. Based on the statements, it can be inferred that there is little education on the treatment of hypertension, which calls for a change in patient counseling methods in the Korean medical community. In this regard, M.D. Ryu shared his opinion as follows.


*What I’m talking about now is the opinions I’ve shared with some doctors around me, so it’s not a generalized answer. But I’m sharing a conversation with doctors around me about why they don’t prescribe exercise to older adults with hypertension. First of all, we rarely learn about specific exercise regimens. I recommend to patients to exercise while looking for medical guidelines or literature, but in reality, there are many difficulties… Because doctors are more interested in medication treatment, it is difficult to give exercise prescriptions. The second reason is that there is a lot of outpatient treatment, so there is less time to devote to each patient. So I have no choice but to spend my time on other more important issues (importance of taking medications, explaining test results, etc.) than exercise. Lastly, most hypertensive patients are mainly middle-aged or elderly. In fact, I know they usually don’t exercise even if I recommend exercising. So, rather, it seems to emphasize the importance of exercise relatively for young hypertensive patients.*
M.D. Ryu’s Statement

The second is the “Lack of expertise in the organization”. As mentioned above, senior welfare centers in Korea (most elderly care-related facilities) are not obligated to hire nurses, doctors, or physical activity specialists. Therefore, it is safe to say that most of the center programs are mainly operated by social workers. For social workers without specialized knowledge, it is practically reasonable to provide the lowest level of physical activity programs when older adults request physical activity programs. At these facilities, it may be more important to prevent injuries or accidents that may occur while older adults with various chronic diseases are physically active, than to improve their health. As a result of the observation of this research team, most of the physical activity programs at senior welfare centers were recreational programs, and many programs provided the lowest level of movement for groups rather than individuals. Accordingly, it is essential that Korean welfare centers need to be innovative, and request an appropriate budget allocation from the government to construct a professional human infrastructure, and secure the social responsibility of health care providers and others [56]. This will allow them to actively intervene in the health and wellness of the elderly.

As shown in the above two themes, it is socio-environmentally difficult for the elderly in Korea to obtain accurate information to protect their health.

### 4.3. Fear of Physical Activity

Through the themes mentioned before, it is possible to better understand why some older adults who participated in this study have a fear of physical activity. The first sub-theme that appeared in this theme is “Trauma or the fear experienced”. One of the participants gave up doing a physical activity because she was afraid of the situation in which her son, who also has hypertension, fell while exercising, and was afraid of the changes in the body that can normally occur during physical activity. In this case, a psychological treatment that can give a message that physical activity is safe may be necessary. Psychological treatment can improve the quality of life [57]. In the second sub-theme, study participants talk about fear of things that have not happened. This can be seen as a kind of “Anticipatory anxiety”. Through news and other indirect experiences, negative aspects of physical activity are recognized, which leads to avoidance of physical activity [57]. Furthermore, if the risk of accidents from physical activity due to other chronic diseases is significantly low, then older adults with hypertension should be encouraged to engage in physical activity by the current guidelines [49]. It has been noted that high-intensity physical activity has more disadvantages than significant advantages for older adults with hypertension, as compared to moderate-intensity physical activity [20]. To help reduce the fear and anxiety of exercising as mentioned by some of the study participants, it is recommended that participate in physical activity if they are accompanied by assistance from family members or close groups. Studies have shown that exercising with family, friends, or in a group can reduce such fears [58].

### 4.4. Rejection of Any Changes

Some of the study participants older adults who participated in this study believed that they were safe and only needed to take the required medication to control their HBP and did not need to do anything else to lower their hypertension with the current blood pressure index, which is controlled by medications, and did not show any other will to improve. This is a reflection of the first sub-theme, “the importance of the current lifestyle.” One study participant, who expressed his thoughts through a picture of a cigarette, stated that he was living his life doing whatever he wanted without worrying too much because his blood pressure was well controlled while he was taking medication. He stated that he did not control drinking alcohol or smoking at all. Furthermore, he stated that he was satisfied with his present life. He prioritized his present life over the future. This seems to be the case for one individual, but in general, the elderly tend to be “current oriented” [59]. This means that there is a stronger tendency to maintain the present way of life rather than considering the big gain in the foreseeable future. This patient seems to be pursuing the lifestyle he wants (alcohol, cigarettes, food, etc.) rather than worrying about the various complications that hypertension can cause. Other studies focusing on older adults have also reported this shown tendency [43]. To change this tendency among older adults, it is important to make them understand the relative advantage among the five most important factors to accept change, according to Rogers’ Diffusion of Innovation [59]. For example, the confidence and satisfaction that physical activity can lead to positive experiences among the older population with hypertension if they are repeatedly given encouragement. The second sub-theme is “Lack of motivation to improve their lives”. Regardless of whether the research participants are satisfied with their current life or not, their unwillingness of them to improve their overall health, as expressed through food, TV, and photographs, is common among older adults [60]. This lack of motivation to change their lifestyle to improve their overall health can also have psychological effects on their caregivers. Therefore, we propose to incorporate specific short-term goals that can motivate this older population with hypertension to engage in physical activity. Having short-term goals regarding physical activity can have a positive effect on their overall health by reducing their reliance on medication only for controlling hypertension. To promote physical activity among an older person with hypertension, this approach should include all those involved with their care, including the older person.

## 5. Limitations

The present study is subject to the following limitation. Although this study pursued more sophisticated sampling through the objective data of the amount of physical activity produced by the accelerometer, it did not examine any medical or physiological causal relationship using the accelerometer data. Moreover, resistant training or water activities are not analyzed by an accelerometer. That is, the data produced in this process cannot be given any meaning other than that it has been sampled. This point is related to the fact that only resistance training is not usually prescribed to hypertensive patients [61]. In addition, this study thoroughly follows the philosophy of qualitative research. Thus, the study conducted a photovoice with 10 research participants. This means that the findings of this study are not externally generalized. However, in a similar environment, this study can provide more precise information. Although photovoice, a method of participatory action research, was used as the research technique, this does not deviate from the framework of this study as a case study. Therefore, the findings of this study cannot be applied to all older adults with hypertension. That is, this study has an inherent limitation in that it cannot be externally generalized. However, it is significant in that the knowledge produced by this study can be applied more precisely to older adults in similar circumstances. This is expressed as an internal generalization by qualitative researchers [40]. Therefore, the accumulation of knowledge produced in this way is essential, which broadens the scope of research.

## 6. Conclusions

Many studies have investigated the benefits of physical activity in older adults with hypertension [8,9,10]. In addition, there have been some studies that directly analyzed the barriers to physical activity in older adults with hypertension [62]. However, most of the studies related to hypertension and physical activity in older adults have been conducted as experimental studies, questionnaire studies, and meta-analyses. There are no studies on how physical activity, which has a very positive effect on older adults with hypertension, is perceived from older adults’ perspective. This study attempted to find out why the majority of older adults with hypertension in Korea do not actively participate in physical activity by using participatory action research called photovoice. Participants in this study expressed their barriers to physical activity that were difficult to predict, difficult to understand from the outside, and invisible through photovoice. This study ultimately found barriers to participation in physical activity directly perceived by older adults with hypertension. If this kind of research continues and results are accumulated, the spectrum of intrinsic generalizations that can be applied in more diverse environments can be broadened. This research team expects that the quality of life of older adults with hypertension will be improved as the follow-up research continues so that they can enjoy a successful aging process. In addition, from a social and national perspective, we expect that the social costs consumed by pathological symptoms of older adults will be reduced through the continuous production of studies similar to this study.

## Figures and Tables

**Figure 1 ijerph-19-14020-f001:**
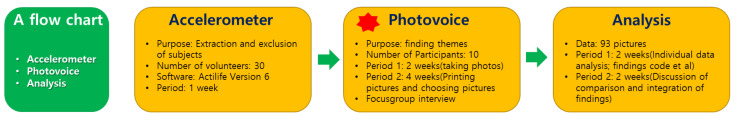
A study Flow Chart.

**Figure 2 ijerph-19-14020-f002:**
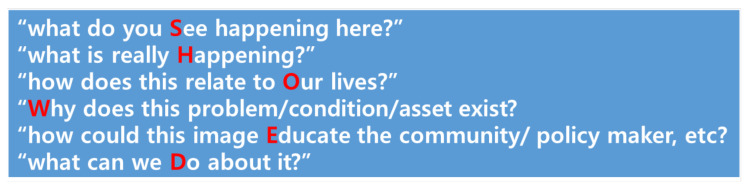
Detail of the SHOWeD Technique.

**Figure 3 ijerph-19-14020-f003:**
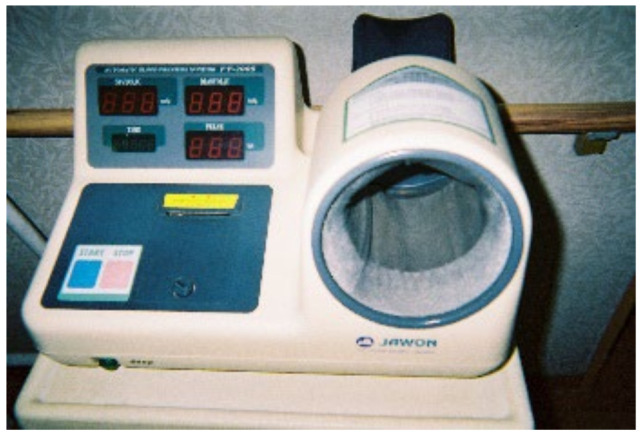
Blood pressure gauge.

**Figure 4 ijerph-19-14020-f004:**
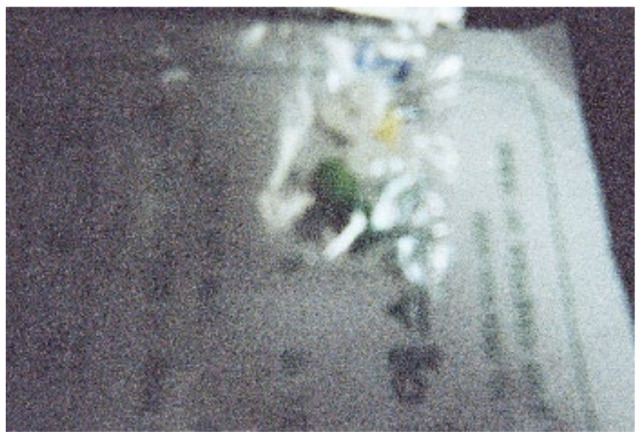
Medication.

**Figure 5 ijerph-19-14020-f005:**
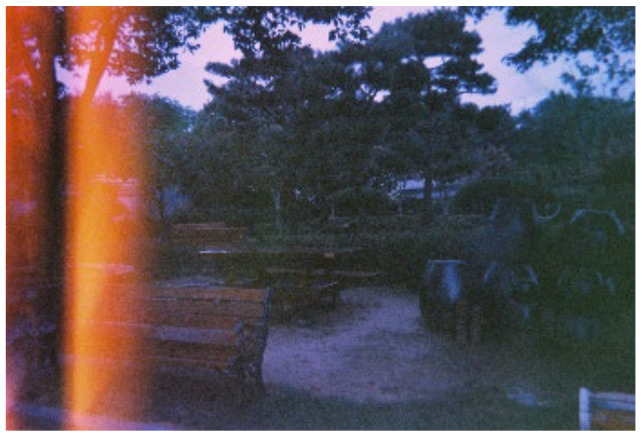
Space in the social welfare center.

**Figure 6 ijerph-19-14020-f006:**
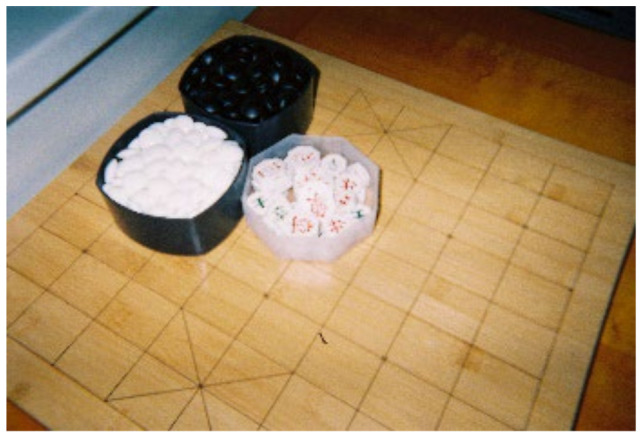
Korean chess.

**Figure 7 ijerph-19-14020-f007:**
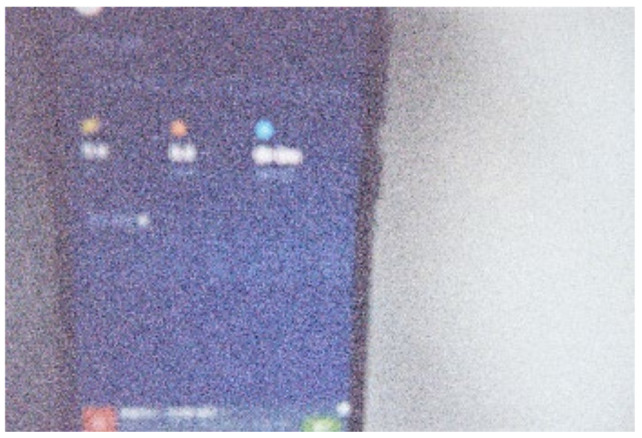
Pedometer.

**Figure 8 ijerph-19-14020-f008:**
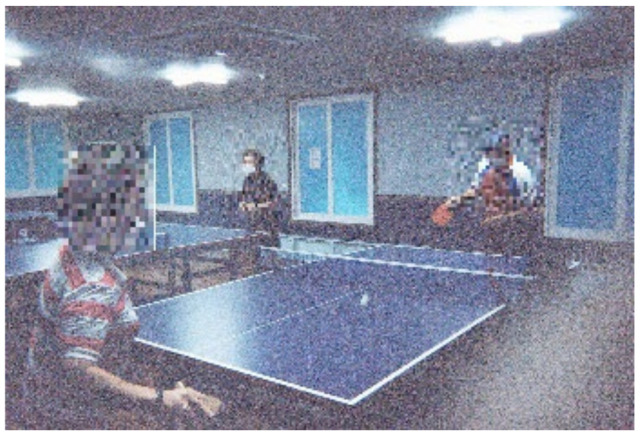
Table tennis court.

**Figure 9 ijerph-19-14020-f009:**
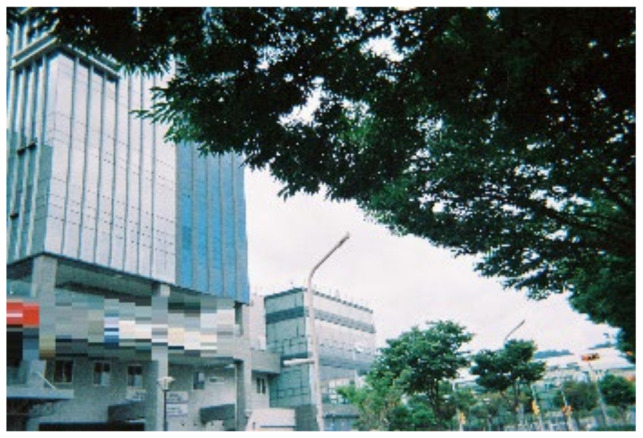
Hospital.

**Figure 10 ijerph-19-14020-f010:**
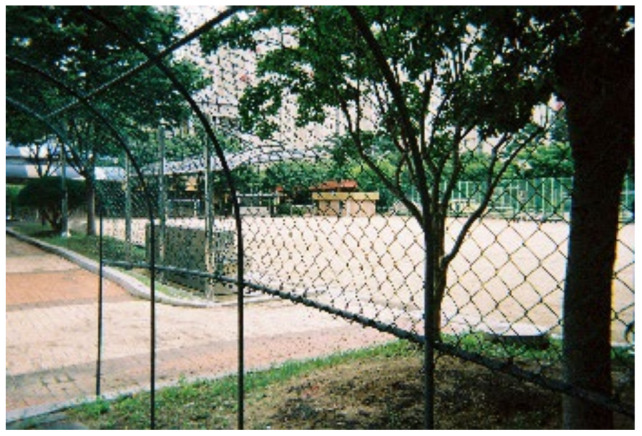
Playground.

**Figure 11 ijerph-19-14020-f011:**
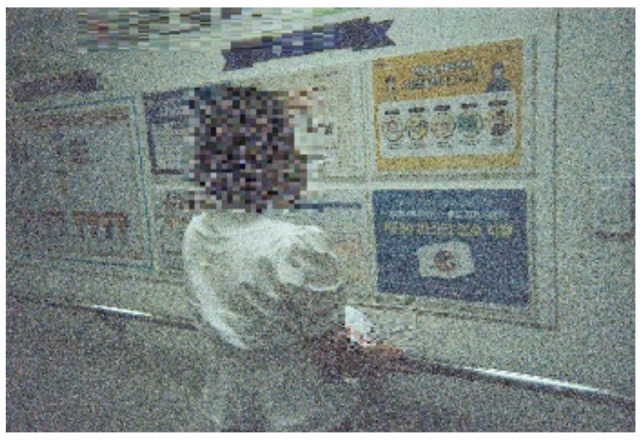
The back of a social worker.

**Figure 12 ijerph-19-14020-f012:**
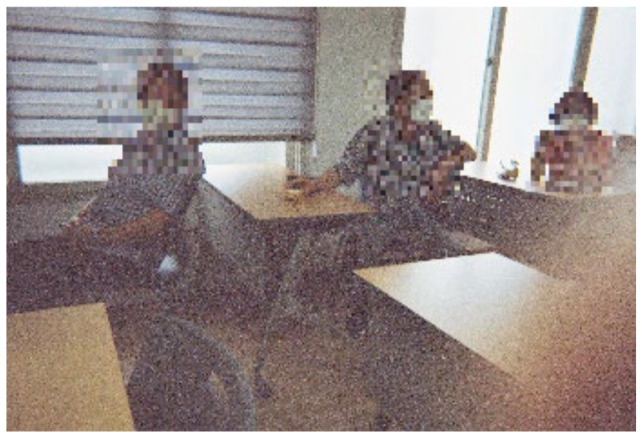
Friends.

**Figure 13 ijerph-19-14020-f013:**
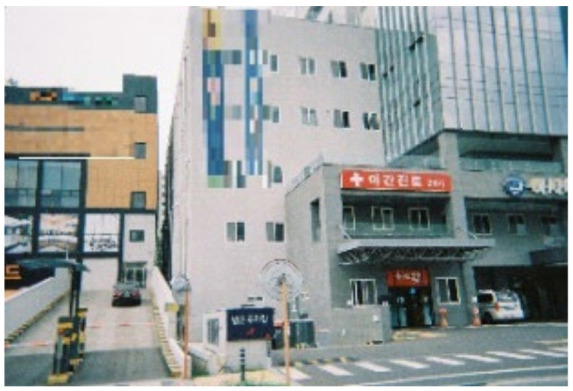
Emergency.

**Figure 14 ijerph-19-14020-f014:**
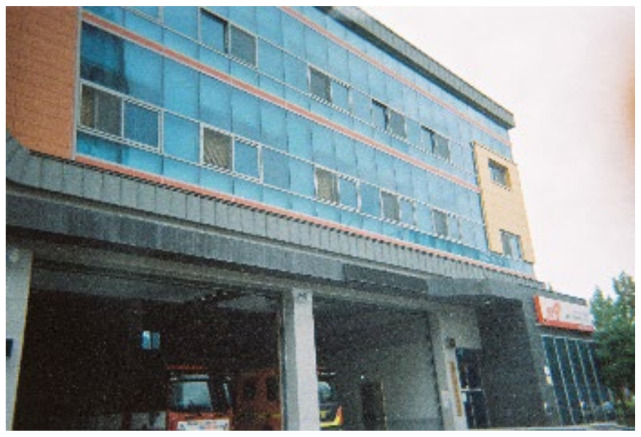
Korean 911(119).

**Figure 15 ijerph-19-14020-f015:**
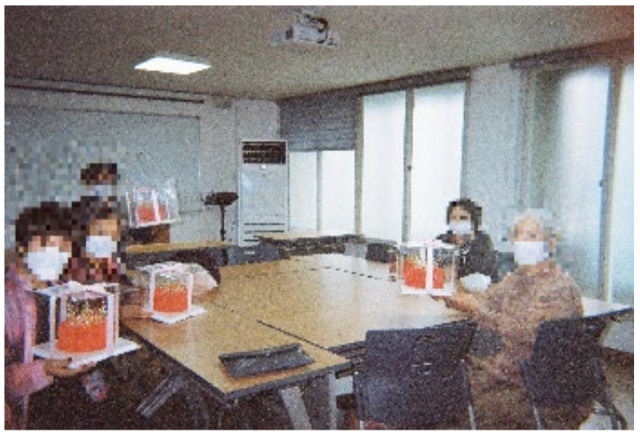
Friends.

**Figure 16 ijerph-19-14020-f016:**
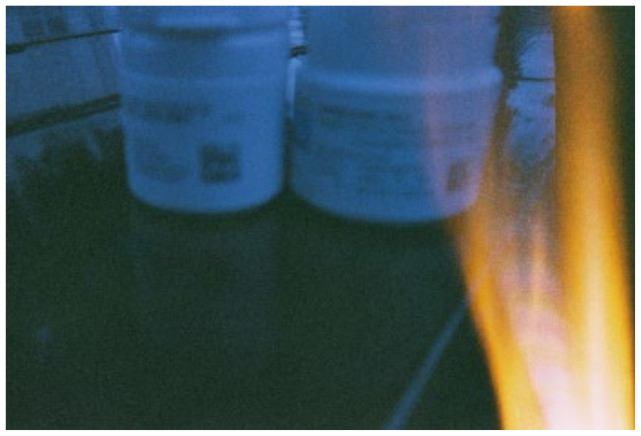
Pill case.

**Figure 17 ijerph-19-14020-f017:**
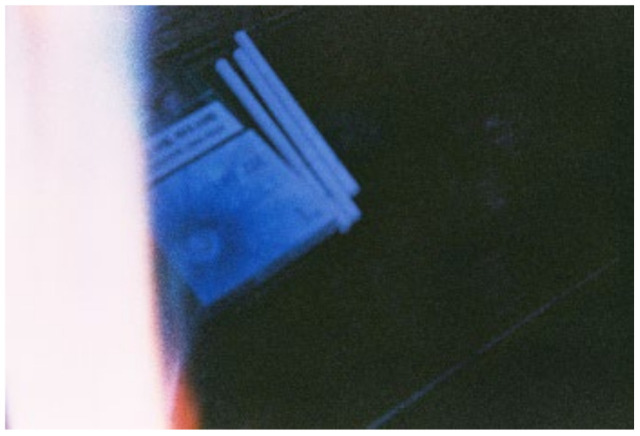
Cigarette.

**Figure 18 ijerph-19-14020-f018:**
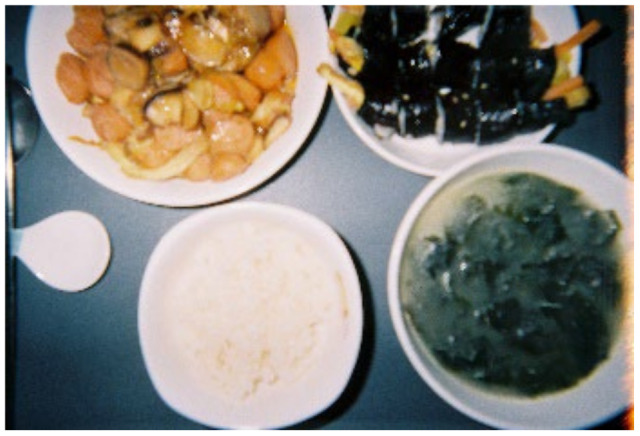
Food.

**Figure 19 ijerph-19-14020-f019:**
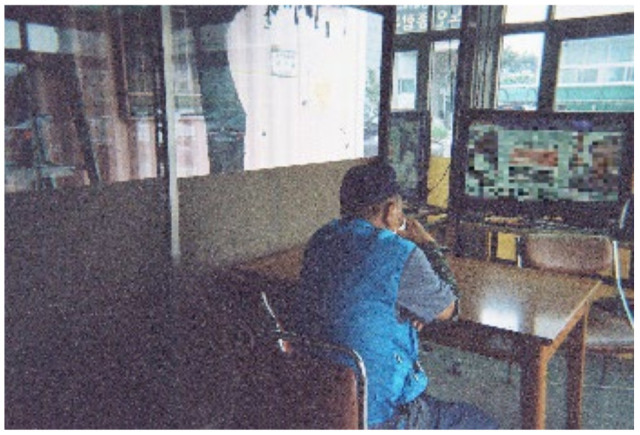
Television.

**Table 1 ijerph-19-14020-t001:** The philosophical background of this study.

Epistemology	Theoretical Perspective	Methodology	Methods
Constructionism	Interpretivism	Case study & Photovoice	Taking Photos, Focus group, Fieldwork, Observation

**Table 2 ijerph-19-14020-t002:** General Characteristics of Participant.

Participant	Gender	Age	Marital Status	Family/ Inmate	Monthly Income ($)	Education Level	Medication	Remark
A	Female	72	Married	Alone	1000 or over	college graduation	Y	-
B	Female	75	Married	Spouse	500 or over	high school graduation	Y	-
C	Female	73	Married	Alone	500 or over	high school graduation	Y	-
D	Male	73	Married	Spouse	1000 or over	college graduation	Y	-
E	Female	73	Married	Relative	1000 or over	high graduation	Y	-
F	Male	74	Married	Alone	1000 or over	high school graduation	Y	-
G	Female	75	Married	Spouse	1500 or over	college graduation	Y	-
H	Female	73	Married	Relative	1500 or over	high school graduation	Y	-
I	Female	72	Married	Relative	1500 or over	high school graduation	Y	-
J	Male	72	Married	Spouse	1000 or over-	high school graduation	Y	-
* Ryu	Male	33	No Info.	No Info.	No Info.	M.D. in public health	-	Researcher (additional interview)

**Table 3 ijerph-19-14020-t003:** Data analysis process.

Stage	Details	Operator
1	Selection of accelerometer participants who meet the criteria for participation in photovoice	All Researchers
2	Finding significant statements with photographs	Kim, Lee, Butler
3	Integration, Organizing, and collecting of significant statements with photographs	Kim, Lee, Butler
4	Classifying categories (ex. Diverse barriers to physical activity) through the classification of important statements	Kim, Lee, Butler
5	Finding main themes and sub-themes in separated categories	All researchers
6	(a) Repeated discussion and reflection of the research team to determine the final theme (b) Holistic analysis based on the overall statement, description, theme, and interpretation (b) Sharing data with research participants for theme confirmation	All researchers & Participants
7	Creating report	Kim

**Table 4 ijerph-19-14020-t004:** Physical Activity Data for a week measured by an accelerometer.

Parti. No	Gender	Age (yr.)	Height (cm)	Weight (kg)	BMI	Physical Activity					M.D.’ Decision Y, N (Another Chronic Disease)	Photovoice Participation (O,X)
						% in Sedentary	% in Light	% in Moderate	Min per Day in Moderate	MET Rate per Week		
1	F	72	150	60	26.6	73.90%	23.34%	2.77%	20.21	1.12	Y	O
2	M	72	168	72	25.5	82.66%	14.73%	2.60%	32.14	1.06	-	X (criteria)
3	F	74	153	45	19.2	81.99%	15.84%	2.17%	26.78	1.09	Y	X (drop-out)
4	F	75	157	62	25.1	81.34%	17.60%	1.06%	13.07	1.04	Y	O
5	F	73	147	47	21.7	79.92%	17.80%	2.27%	19.42	1.03	Y	O
6	M	71	170	60	20.7	85.71%	14.20%	0.09%	1.07	1.00	N (arthritis)	X (judgment)
7	M	73	167	65	23.3	83.17%	16.08%	0.75%	9.28	1.04	Y	O
8	F	73	153	60	25.6	79.35%	15.49%	4.83%	59.57	1.16	-	X (criteria)
9	F	70	169	69	24.1	66.75%	28.51%	4.75%	58.57	1.21	-	X (criteria)
10	M	75	158	69	27.6	89.53%	10.35%	0.12%	1.50	1.00	N (depression)	X (judgment)
11	M	73	170	80	27.6	86.22%	13.63%	0.14%	1.78	1.01	Y	O
12	M	75	168	69	24.4	80.57%	15.24%	4.19%	51.71	1.15	-	X (criteria)
13	F	74	149	58	26.1	92.85%	6.67%	0.45%	5.57	1.02	Y	O
14	F	71	155	55	22.8	70.34%	24.82%	4.83%	59.57	1.16	-	X (criteria)
15	F	74	148	67	30.5	78.28%	20.61%	1.10%	13.57	1.03	N (obesity)	X
16	F	75	164	62	23.0	85.39%	14.33%	0.28%	3.42	1.02	Y	O
17	F	72	152	59	25.5	84.01%	13.10%	2.89%	35.71	1.06	-	X (criteria)
18	F	73	147	51	23.6	86.74%	12.52%	0.74%	9.14	1.02	Y	O
19	M	72	169	69	24.1	88.50%	11.15%	0.36%	4.42	1.00	Y	O
20	F	70	157	70	28.3	86.36%	11.56%	2.08%	25.64	1.08	Y	X (judgment)
21	F	73	147	50	23.1	79.84%	19.46%	0.71%	8.71	1.06	N (stroke)	X (judgment)
22	F	72	150	45	20.0	82.95%	15.72%	1.33%	16.35	1.07	Y	O
23	M	72	152	51	22.0	87.14%	12.51%	0.35%	4.28	1.02	N (stomach cancer)	X (judgment)
24	F	75	159	70	27.6	84.46%	12.19%	3.29%	40.57	1.10	-	X (criteria)
25	M	74	168	74	26.2	82.80%	14.97%	2.24%	27.64	1.11	Y	X (drop-out)
**Mean**		72.95	158	62	24.6	82.57%	15.57%	1.84%	21.99	1.07		

**Table 5 ijerph-19-14020-t005:** Presentation of the obtained findings.

Main Theme	Sub-Themes	Title of Photos
Health illiteracy	- Medication dependency	- blood pressure gauge - medication
	- A negative reference group	- space in the social welfare center - Korean chess
	- Misperception of appropriate physical activity	- pedometer - table tennis court
Distortion of health information	- Inappropriate health information from M.D.	- hospital- playground
	- Lack of expertise in the organization.	- the back of a social worker -friends
Fear of physical activity	- Trauma or The fear experienced	- emergency - Korean 911(119)
	- Anticipatory anxiety	- friends - pill case
Rejection of any changes in life	- The importance of the current lifestyle	- cigarette
	- Rejection of any changes	- food - television

## Data Availability

The data presented in this study are available on request from the corresponding author. The data are not publicly available to protect respondent privacy.

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
