# Peer review of "Exploring Perceived Barriers to Physical Activity in Korean Older Patients with Hypertension: Photovoice Inquiry"

_ijerph, 2022, doi:10.3390/ijerph192114020_

Round 1
Reviewer 1 Report
The Authors submitted an interesting paper about physical activity in older Koreans with Hypertension.
Line 38-40: stroke and brain infarction are the same disease, so brain infarction should be eliminated from the list. Cushing syndrome is NOT a consequence of hypertension, being due to excessive corticosteroid production. Heart attacks and Myocardial infaction are the same diseas, so heart attacks should be eliminated. What disease is "cytoma"? Necrosis of what organ/System?
Line 42: "access" should read "assess"
Line 44-45: Medication treatment should read "pharmacological treatment"
Line 107: what is the difference between financial costs and economic costs? Are they not the same?
Methods: how were the participants chosen? What is the risk of bias? Why a random sampling was not used? The "photovoice" should be explained better, because it's not clear what it is. What people composed the focus groups?
Line 374-376: how is reference 47 (relative to dogs and OCULAR hypertension) useful or meaningful in a paper about arterial hypertension in humans? Where in the listed references is said that hypertension is a risk factor for cancer? Why is therapeuthic compliance (ie taking pills as directed by a physician) regarded as "medication addiction"?
Reviewer 2 Report
The article “Exploring Perceived Barriers to Physical Activity in Korean 2 Older Patients with Hypertension: Photovoice Inquiry” by Lee et al., highlight the various factors which prevent the physical activity of old patients suffering from Hypertension. The article is well written and consider to be published in the reputed journal with minor revision.
11. The number of participants is just 10 which is too low for this kind of study. It would be better to lower the older age from 70 to 65 years and increase the number of participants.
22. Authors used so many pictures like- blood pressure gauge, medication, space in the social welfare center, pedometer etc. etc. I do think only a few pictures are enough for the study and there is no need to include so many pictures.
32. Discussion part is too long. It should be very precise and focused.
Reviewer 3 Report
Lee et. al provide an interesting study describing causes of physical inactivity in elderly persons with hypertension.
While the topic is very interesting, there are some points, which should be addressed:
1. For persons not familiar with the photovoice technique -it would be great, if this could be clarified better in the abstract
2. The study group is pretty small, please discuss this as a limitation - also in the abstract. Is there a possibility to include more participants? If not, I would rather term the observations as pilot.
3. Please provide careful language correction by a native speaker - I have noticed some word repetitions throughout the text.
